# Meat Fatty Acid Composition of Wild Boars Hunted in Romania in Relationship to Gender and Age-Class

**DOI:** 10.3390/ani12070810

**Published:** 2022-03-22

**Authors:** Marius Mihai Ciobanu, Alina Narcisa Postolache, Florin Daniel Lipşa, Mugurel Munteanu, Roxana Nicoleta Rațu, Otilia Cristina Murariu, Paul Corneliu Boișteanu

**Affiliations:** 1Faculty of Agriculture, “Ion Ionescu de la Brad” University of Life Sciences, M. Sadoveanu Alley, No. 3, 700490 Iasi, Romania; marius.ciobanu@uaiasi.ro (M.M.C.); roxana.ratu@uaiasi.ro (R.N.R.); otiliamurariu@uaiasi.ro (O.C.M.); 2Research and Development Station for Cattle Breeding Dancu, 707252 Iasi, Romania; narcisa.postolache@gmail.com; 3Faculty of Animal and Food Resources Engineering, “Ion Ionescu de la Brad” University of Life Sciences, M. Sadoveanu Alley, No. 8, 700490 Iasi, Romania; munteanumugur@uaiasi.ro (M.M.); paulb@uaiasi.ro (P.C.B.)

**Keywords:** wild boar meat, carcass value, fatty acid profile, effect of gender and age-class

## Abstract

**Simple Summary:**

Game meat has a good nutritional profile; wild boar, in particular, has lower fat, high protein, and higher iron content than meat from other domesticated animals. The present study aimed to assess the carcass characteristics and chemical composition and also to investigate the effect of gender and age on the intramuscular fatty acid profile of wild boar meat harvested during the 2010–2017 winter hunting seasons in the Frasin District hunting area (Romania). The age-class factor influenced the carcass traits (*p* ≤ 0.05), but not the yield, although overall, adults were heavier and had a better yield compared to the younger specimens (77.36 vs. 74.79 kg). The differences induced by gender for carcass traits and dressing yield were present but not statistically significant (*p* > 0.05). For the chemical composition of wild boar meat, the percentages of protein, fat, ash, and water did not present significant differences (*p* > 0.05) within the factors applied (gender, age-class). Regarding the studied effects, results showed that the wild boars’ age-class significantly influenced (*p* ≤ 0.001) the amount of meat total fat, with the adults having the highest fat content than subadults (7.60% vs. 4.52%). As for the fatty acid profile, a high ratio of PUFA/SFA (P/S: polyunsaturated fatty acids/saturated fatty acids) and n-6/n-3 PUFA was present in the meat, showing good functional properties for human health.

**Abstract:**

The present study evaluates the influence of gender and age-class on the carcass traits, meat chemical composition, and fatty acid profiles of randomly hunt-harvested wild boars from Romania’s Frasin District hunting area, which is an important part of the stock density management strategy. Only 76 wild boars were chosen to participate in the experimental design from a total of 94 wild boars hunted over seven winter seasons, based on two main criteria: right carcass part intact and animal health status (suitable for human consumption) as determined by the veterinary health notice issued by the Food Safety Authority, with all wild boars being evaluated before slaughtering. After sanitary inspection and within the first 24 h postmortem, meat samples were taken in duplicate from the right side of wild boar carcasses (m. *Biceps femoris*). The aim of this study was to compare the carcass characteristics, chemical composition, and fatty acid profile of wild boar *Biceps femoris* muscle obtained from the right side of carcasses, as well as the effects of gender, age-class, and their interaction on these traits, in a context where venison is frequently regarded as organic food by consumers due to the natural habitat in which wild animals live. The age of the animals had an effect on the postmortem undressed warm and cold carcass weights (*p* ≤ 0.05). Considering the proximal chemical composition, only the intramuscular fat (IMF) content was higher in adult meat samples compared to young animals. C18:1 n-9 was the most abundant fatty acid in wild boar meat, followed by C16:0, C18:2 n-6, and C18:0. For IMF content, gender, age class, and their interaction influenced mainly MUFA fatty acids, in particular C16:1 n-9, C20:1 n-9 and C22:1 n-9, and also PUFA fatty acids C18:3 n-6, C20:3 n-3, C20:4 n-6, C20:5 n-3, C22:4 n-6, and C22:5 n-6. The appropriate amounts of individual fatty acids in the muscles of the wild boars translate into a ratio of P/S acids that promotes health as a bio-alternative resource of meat.

## 1. Introduction

Consumers today are encouraged to look for alternative meat types due to the changes in their eating habits and the increased disease risk associated with conventional farm meat [1,2] For these reasons, game meat has grown in popularity, despite the fact that consumption of game meat has traditionally been higher in mountainous areas. There is a link between these areas and hunting activity, as harvested animals are consumed by hunters and their families, as well as those who have a certain degree of knowledge and are familiar with game meat products [3,4]. The latest survey on the consumption of game meat in Europe showed higher consumption in the South-East region than in Central Europe [5]. Men and older consumers eat more game meat than women and younger consumers due to its nutritional attributes, organic characteristics, sensory properties, and well-known health-related benefits [6]. This positive trend has resulted from a combination of factors, including an increase in wild boar populations in Europe due to their adaptability and fertility [7,8,9,10,11], the importance of animal welfare-related to the development of raising practices as close to their natural environment as possible [12], consumer interest in organic game meat [13], and its quality, particularly in terms of health-related fat quality meat indicators [12]. Consumers’ choice for game meat derives from a series of factors that include availability, price, and quality. The quality concept from the consumer’s point of view includes microbiological safety, nutritional value, and the sensory profile in terms of color, aroma, flavor, taste, texture, and tenderness [14].

It is generally known that game meat has a good nutrient composition, wild boar, in particular, has a low-fat content, a high protein content, and a higher iron content compared to pork or beef meat [15,16,17]. Game meat is considered to be healthier than meat from domesticated animals because of its lipid content, fatty acid profile, and nutritious protein [18,19].

Intramuscular fat is well known to contribute to meat’s sensory attributes [12]. The gender of the animal influences fat deposition, with an important impact on the flavor of the meat. Females assimilate protein differently and tend to accumulate more fat than males, and have a higher percentage of fat at any chronological age [20]. Despite the fact that numerous studies on the fatty acid composition of wild boar meat have been conducted in various parts of the world [21,22,23,24], no studies on wild boar meat hunted in Romania and used as a raw material for industrial processors have been done.

Plants (grasses, leaves, roots, bushes, seeds, forest fruits) and, less frequently, bird eggs, snails, insects, earthworms, larvae, and beetles are the primary food sources for wild boars in their natural habitats [25,26]. Due to the natural habitat in which wild animals thrive, venison is frequently referred to as organic food. In light of this, the objectives of this research were to investigate the carcass characteristics, chemical composition, and fatty acid profile of wild boar *Biceps femoris* muscle and also to analyze the effects of gender, age-class, and their interaction on the mentioned traits.

## 2. Materials and Methods

### 2.1. Animals and Muscle Samples

The study included 76 wild boars (*Sus scrofa ferus*), which were harvested under Romania’s national game, hunting, and wildlife protection legislation [27] during the 2010–2017 hunting seasons (winter), with the purpose of stock density control in the Frasin District hunting area. At the beginning of this study, the livestock spring population of wild boars had an estimated value of 95 animals (Table 1).

Since the availability of feed is limited in cold winters, between 1 November and 31 March of each year, additional feed in the amount of 86 kg/animal/year was provided in concentrated (cereals and seeds) and juicy forms (fresh fruits, beets, turnips, potatoes, and carrots).

Animals were shot during the morning period, and the sanitary inspection was carried out outside during the afternoon (at the evisceration checkpoint), with an average external temperature between −5 °C and −15 °C. Those shot in the upper right thigh were eliminated from the study.

The carcass parameters for all animals were determined using a calibrated scale with a limit of 150 kg, as follows: Warm carcass weight represents the weight of animals shot without organs and kidneys 5 h after shooting; cold carcass weight represents the weight of animals shot without organs, and kidneys 24 h after slaughtering, and dressing percentage represents the cold carcass as a percentage of its undressed weight. The animal’s age was calculated using tooth eruption, which takes over 26 months to complete, replacement, and wear patterns [28,29].

The samples used to determine the proximate composition and total lipid (intramuscular fat) fatty acid composition was *m. Biceps femoris* (duplicate samples/muscle) obtained from the right side of wild boar half carcasses. After the sanitary inspection [30] in the first 24 h postmortem, muscle samples were collected, identified according to gender and age-class, sealed in sterile bags and transported to the laboratory under refrigerated conditions (0–4 °C), trimmed of connective and adipose tissues, vacuum packed, and frozen at −20 °C until analysis.

### 2.2. Proximate and Fatty Acid Analysis 

The samples were thawed overnight, hacked, and homogenized prior to analysis. According to AOAC [31], the nutrient content was analyzed for moisture (AOAC 950.46), ash (mineralization at 550 °C for 16 h according to AOAC 920.153), fat (extraction under hot treatment influence, with petroleum ether as a solvent, in a Soxtherm 416 equipment—AOAC 960.39) and protein content (Kjeldahl method—AOAC 928.08).

### 2.3. Fatty Acid Analysis 

Gas-liquid chromatography was used to determine the concentration of individual fatty acids in two extracts from all samples [32]. As an internal standard, 3–5 mg of nonadecanoic acid (C19:0; 1.5–2.5 g) was added to the sample (1.5–2.5 g) (IS). Folch et al. [33] described the extraction of lipids from meat samples with a mixture of chloroform and methanol (2:1 *v/v*). Following that, the lipid extracts were trans-esterified with methylene to produce fatty acid methyl esters (FAME). A CarloErba 5300 mega-series GC gas chromatograph equipped with a flame ionization detector suited for a fused-silica Omegawax 320 capillary column type SP-2380 was used to separate and quantify FAME (60 m × 0.25 mm internal diameter × 0.20 m film thickness, Supelco Inc., Bellafonte, PA, USA). The following were the chromatographic operating conditions: the initial column oven temperature of 160 °C was programmed to rise at a rate of 4 °C/min up to 250 °C, with a 45-min run time. Temperatures in the injection and detector ports were kept at 250 °C and 260 °C, respectively. Helium was used as the carrier gas, with a flow rate of 1.2 mL/min and a splitting ratio of 1:20. By comparing the retention times of the standard fatty acid methyl esters, the peaks were identified.

### 2.4. Statistical Analysis

All statistical analysis was carried out using the SPSS v.20 software package (SPSS Inc., Chicago, IL, USA). The carcass characteristics were subjected to one-way analysis of variance with gender (males and females) and age-class (sub-adults and adults) as fixed factors. A general linear model (GLM) was used to examine the fatty acid profiles and chemical composition, using gender (males and females), age class (sub-adults and adults), and their interaction as fixed factors. As corrected confounders, carcass weight and muscle fat content were included. By evaluating the association between the various FA and condensing them into meaningful components, principal component analysis (PCA) was utilized to analyze and comprehend the variability of wild boar meat composition (PCs).

## 3. Results and Discussion

To our knowledge, this study is the first comprehensive assessment of the nutritional value of Romanian hunted wild boar muscle lipids. The descriptors for the obtained wild boar’s carcass are shown in Table 2. The undressed weight, warm carcass weight, and cold carcass weight of wild boar were higher on males than females, adults vs. sub-adults, with significant differences (*p* ≤ 0.05). Our findings are in accordance with early preliminary results for the differences found in dressed weight between males (65–108 kg) and females (50–80 kg) of 3–4 years old [34] or other data research summarized by Pedone et al. [35], as cited in Sales and Kotrba’s [36] review of the effects of animal age and gender on carcass weights of wild boars hunted in different European countries. These results support the findings of other authors [15,37,38,39,40], who state that the meat quality of wild boars is determined by the season, available feed, as well as the animals’ living conditions, and sexual activity.

The proximate composition of wild boar meat is shown in Table 3. The percentage of protein, ash, and water was not influenced by gender (*p* > 0.05). Only intramuscular fat content was influenced by age-class, with adults having a significantly (*p* ≤ 0.001) higher content than subadults, a result that can often be explained by age, feed availability amount, and sexual activity [36].

The chemical composition reported by Marsico et al. [41], Strazdina et al. [10] and Lazăr et al. [42] was similar to that found in the present study, in terms of moisture (69–70.5%), protein (20.16–22.92%) and ash content (1.15–1.30%). The influence of gender and age-class on these parameters was not significant (*p* > 0.05). Our results (over 4.5%) were higher than those reported by the above-mentioned authors but close to the values obtained by Reka et al. [43] (4.3–6.4% IMF), and in accordance with previous intermediate studies by Postolache et al. [44] (2.65–5.22%) for wild boars hunted in Romania. The results for IMF showed that age-class had a significant impact (*p* ≤ 0.001), with mean values ranging from 4.52% for subadults to 7.60% for adults.

The literature points out that animal diets are reflected in their meat chemical composition, which depends on the season and animal food availability [24]. Moreover, IMF content is influenced by age-class; Neethling et al. [45] reported that IMF content increases with age, being the last tissue to mature. IMF content differences are mainly a result of animal age and various animal feed intakes: 1.23–4.27% [44], 3.5–5.2% [46], 1.95% [47] or 4.27–14.2% [26].

Table 4 presents data on the fatty acid profile of wild boar intramuscular fat across a number of 24 fatty acids with varying saturation levels and chain lengths, as well as their health-related lipid indices. Monounsaturated fatty acids (MUFA), which accounted for 41.8–44.3% of the total IMF, were the most prevalent lipid component in wild boar meat, according to total fat content measurements. C18:1 n-9 was the most major MUFA group, accounting for 79.52% of all MUFA. With average values between 32.7% and 34.30% of total intramuscular FA, the saturated FA (SFA) fraction was the second most important group. The most prominent C16:0 and C18:0 were identified, accounting for 64.15% and 33.4% of total SFA, respectively. All of the other SFAs were insignificant. The polyunsaturated fatty acids (PUFA) computed average values ranged from 17.12% to 19.19% of total intramuscular FA, making them the third most important fraction of total FA in terms of percentage importance. With an average value of 74.44% of total PUFA, C18:2 n-6 was the most prevalent in terms of amount. With an average of 10.61% of total PUFA, C20:4 n-6 came in second. No effect (*p* > 0.05) of gender, age-class, or their interactions was found for SFA, MUFA, PUFA, and Fatty Acid Indices.

Most influenced (*p* ≤ 0.05; *p* ≤ 0.01; *p* ≤ 0.001) fatty acid composition typology by gender, age group or their interactions was in case of PUFA content. IMF of females displayed significantly greater percentages for C15:0, C22:0, C20:1 n-9, C22:1 n-9, C18:2 n-6, C18:3 n-6 and lower percentages of C16:1 n-9, C20:3 n-3, C20:4 n-6, C20:5 n-3, C22:4 n-6, C22:5 n-6 in comparacy to males. Regarding age effect, adults had a higher content in C14:0, C15:0 (*p* ≤ 0.05), C20:1 n-9, C22:1 n-9 (*p* ≤ 0.01) and lower in C16:1 n-9, C18:3 n-6, C22:5 n-6 (*p* ≤ 0.05) than sub-adults.

Based on the study results, the fatty acid composition of wild boar muscle tissues showed that increasing animal maturity reduces the proportion of C22:0, C16:1 n-9, C18:3 n-6, and C22:5 n-6, findings that differ from those reported by Razmaitė et al. [25] for the same subject, but for eicosapentaenoic acid (C20:5 n-3), margaric (C17:0), heptadecenoic (C17:1), linolenic (C18:3 n-3), arachidonic (C20:4 n-6), and docosapentaenoic (C22:5 n-3) acid composition of wild boar muscles, organs, or adipose tissue.

The fatty acid composition of our samples was aligned with the results of other studies [10,25,48], where C18:1 n-9 (oleic) was the most abundant fatty acid, followed by C16:0, C18:2 n-6, and C18:0. The same decreasing order of concentration was found in wild boar meat with different feeding regimes (natural feed, supplementary feed, complete diet) [26], age, or gender [44].

The fatty acid composition of wild boar meat is a reflection of their diet [49,50]. These differences were reported in terms of supplementary feed influence on SFA content in wild boars (m. serratus anterior) hunted during the cold season in Hungary. The animals with a controlled complete diet had a lower SFA content (32.91% of total fatty acids) compared to boars that have access to supplementary feed (39.13%) or without access (34.72%) [26].

The nutritional quality of wild boar’s IMF was assessed through the PUFA: SFA acid ratio (P/S) and the n-6/n-3 PUFA ratio. For a healthy human diet, the recommended ratio for P/S is 0.4 or higher [51], and the ratio for n-6/n-3 PUFA was less than 4 (WHO). Due to the content of C18: 2 n-6, the P/S ratio (0.50–0.59) in our wild boar meat was slightly higher than that obtained by Razmaitė et al. [25] (0.27–0.53) in meat from wild boars hunted in Lithuania, but similar to the values obtained by Quaresma et al. [48] (0.52–0.60) and Strazdina et al. [10] (0.50) in meat from wild boars hunted in Portugal. These results highlighted that the lipid composition of wild boar raw meat provides a source of fatty acids (AA, EPA, DHA, and LA) with functional properties for human health and should be included in a balanced diet.

The principal component analysis was used to create a simplified set of parameters that account for the majority of the variability in the original data. The retention of the principal components was performed according to PCs eigenvalues > 1. Six principal components that fit these criteria explained 95.07% of the wild boar meat fatty acids content total variance. Figure 1 displays a plot of the scores for the first two principal components (PC1 and PC2), which accounted for 60.62% of the overall variation. PC1 defined 37.52% of the total variance and for the most part was associated positively with C16:1 n-9 (0.873), C20:3 n-3 (0.801), C20:4 n-6 (0.850), C20:5 n-3 (0.678), and C22:4 n-6 (0.884), and negatively with C15:0 (−0.802), C18:0 (−0.732), C22:0 (−0.694), C20:1 n-9 (−0.858), C18:2 n-6 (−0.671), C18:3 n-6 (−0.635), and C20:2 (−0.684). PC2 explained 23% of the total variance and it was positively correlated with C18:1 n-7 (0.520), C22:1 n-9 (0.718), C22:5 n-3 (0.650), and C22:6 n-3 (0.761), and negatively with C14:0 (−0.884) and C22:5 n-6 (−0.670).

Principal Component Analysis (PCA) also identifies clear differences and affinities in the composition of fatty acids between samples and groups of animals, respectively (Figure 2).

## 4. Conclusions

According to the above-mentioned findings, the primary unprocessed traits of wild boar carcasses were affected by gender and age-class (*p* ≤ 0.05), with adults, particularly males, weighing more than subadults and females. In terms of chemical composition, total fat content was significantly (*p* ≤ 0.001) influenced by the age-class of the animals, with adults having fatter meat than younger animals. Furthermore, gender and age-class influenced the content of some individual MUFA (C16:1 n-9, C20:1 n-9, C22:1 n-9) and PUFA (C18:3 n-6, C22:5 n-6) fatty acids in the context where total MUFA content was highest in males and adults and PUFA content was highest in females and subadult meat samples.

In summary, these findings suggest that the low intramuscular fat content of wild boar meat is the most advantageous feature that can be used to promote the species. This is due to the fact that sufficient amounts of individual fatty acids in wild boar muscle translate into a health-promoting P/S acid ratio, making it a bio alternative protein resource.

## Figures and Tables

**Figure 1 animals-12-00810-f001:**
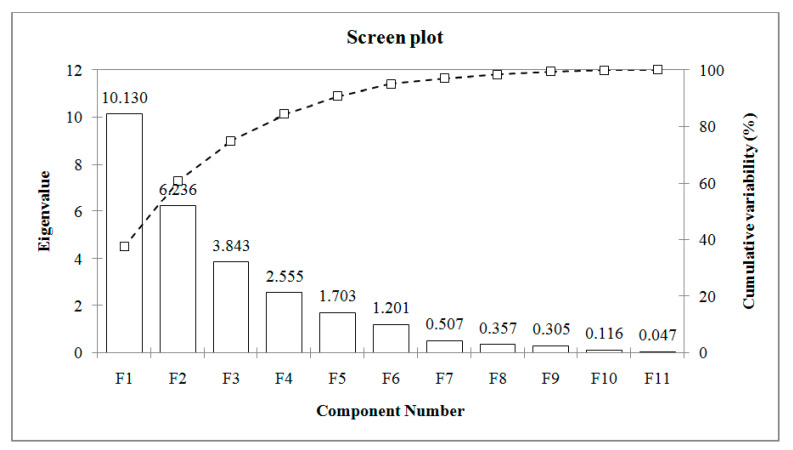
PCA screen plots of fatty acids on all samples and variables.

**Figure 2 animals-12-00810-f002:**
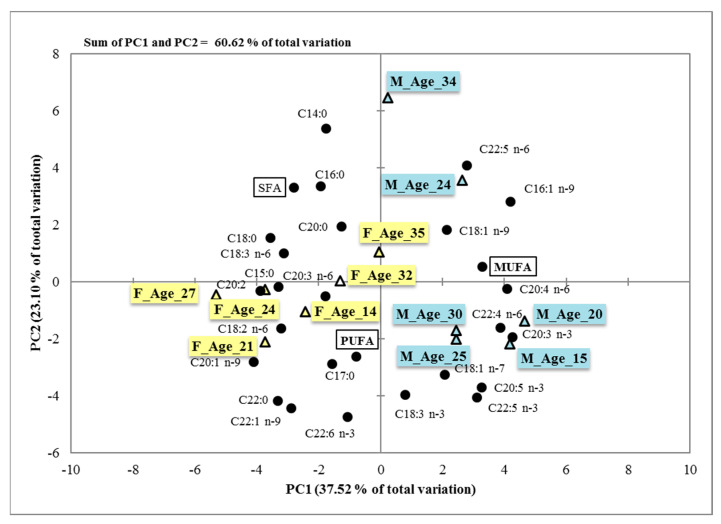
PCA factor-scores biplots, explaining the two components score (PC1 and PC2) for the samples of wild boar’s meat and display grouping similarities between samples according to their fatty acids concentration. SFA = Saturated Fatty Acids, MUFA = Monounsaturated Fatty Acids, PUFA: Polyunsaturated Fatty Acids; F_Age (14, 21, 24, 27, 32, 35): samples from female wild boars aged 14 to 35 months; M_Age (15, 20, 24, 25, 30, 34): samples taken from wild boar males aged 15 to 34 months.

**Table 1 animals-12-00810-t001:** Wild boar population dynamics in the Frasin Forest District.

Wild Boar Population	Hunting Seasons						
	2010/2011	2011/2012	2012/2013	2013/2014	2014/2015	2015/2016	2016/2017
Spring livestock (no.)	95	100	98	98	101	97	104
Hunting quota (no.)	15	15	17	19	20	21	22
Hunted animals (no.)	15	10	11	13	14	15	16

**Table 2 animals-12-00810-t002:** Wild boar’s carcass distribution (mean ± SEM).

Trait	Gender		Age-Class ^e^		Effect		
	Male (n = 27)	Female (n = 49)	Subadults ^A^ (n = 39)	Adults ^B^ (n = 37)	Gender	Age-Class	Gender × Age-Class
Undressed weight a (kg)	92.77 ± 9.36	76.76 ± 4.19	71.89 ± 3.77	97.64 ± 7.10	0.03	0.05	0.18
Warm carcass weight b (kg)	74.90 ± 8.29	59.99 ± 3.51	56.49 ± 3.87	78.40 ± 6.35	0.03	0.03	0.13
Cold carcass weight c (kg)	72.01 ± 8.10	57.70 ± 3.52	53.96 ± 3.69	75.75 ± 6.13	0.03	0.04	0.16
Dressing percentage d (%)	77.09 ± 1.33	75.05 ± 0.57	74.79 ± 1.13	77.36 ± 0.75	0.16	0.88	0.73

^a^ weight of the freshly shot animal; ^b^ without organs but with kidneys, 5 h after shooting; ^c^ without organs but with kidneys, 24 h after slaughtering; ^d^ cold carcass as a proportion of undressed weight; ^e^ animal age was estimated based on tooth eruption, replacement and wear pattern; ^A^ animals with age ranging from 13–24 months; ^B^ animals with age older than 24 months; SEM: standard error of the mean; ns: *p* > 0.05; significant: *p* ≤ 0.05.

**Table 3 animals-12-00810-t003:** Chemical composition (mean ± SEM) of *Biceps femoris* muscle from wild boars hunted in Romania.

Traits	Gender		Age-Class ^e^		Effect		
	Male (*n* = 27)	Female (*n* = 49)	Subadults ^A^ (*n* = 39)	Adults ^B^ (*n* = 37)	Gender	Age-Class	Gender × Age-Class
Moisture (%)	64.59 ± 2.12	61.83 ± 1.00	61.86 ± 1.75	64.55 ± 1.58	0.262	0.274	0.315
Protein (%)	22.76 ± 0.56	22.00 ± 0.53	22.78 ± 0.59	21.99 ± 0.50	0.365	0.347	0.494
Fat (%)	5.56 ± 0.75	6.55 ± 0.82	4.52 ± 0.23	7.60 ± 0.58	0.129	0.001	0.525
Ash (%)	1.14 ± 0.05	1.06 ± 0.17	0.99 ± 0.11	1.20 ± 0.13	0.676	0.243	0.188

^A^ animals with age ranging from 13–24 months; ^B^ animals with age older than 24 months; ^e^ animal age was estimated based on tooth eruption, replacement, and wear patterns; SEM: standard error of the mean; ns: *p* > 0.05; significant: *p* ≤ 0.001.

**Table 4 animals-12-00810-t004:** Effect of gender and age-class on the fatty acid composition (% of total fatty acids, mean ± SEM) of M. *Biceps femoris*, corrected for carcass weight and muscle fat content component.

	Gender		Age-class ^e^		Effect		
	Male (n = 27)	Female (n = 49)	Subadults ^A^ (n = 39)	Adults ^B^ (n = 37)	Gender	Age-Class	Gender × Age-Class
C14:0	0.24 ± 0.161	0.31 ± 0.303	0.27 ± 0.105	0.28 ± 0.128	0.197	**0.017**	0.964
C15:0	0.01 ± 0.007	0.04 ± 0.010	0.03 ± 0.011	0.02 ± 0.013	0.033	0.987	0.864
C16:0	21.44 ± 0.501	21.62 ± 0.630	20.93 ± 0.215	22.13 ± 0.736	0.745	0.976	0.592
C17:0	0.23 ± 0.010	0.24 ± 0.009	0.23 ± 0.012	0.23 ± 0.008	0.605	0.298	0.229
C18:0	10.89 ± 0.192	11.50 ± 0.168	10.99 ± 0.284	11.39 ± 0.190	0.065	0.761	0.643
C20:0	0.02 ± 0.008	0.06 ± 0.017	0.04 ± 0.010	0.04 ± 0.020	0.052	0.278	0.404
C22:0	0.12 ± 0.029	0.23 ± 0.020	0.18 ± 0.040	0.18 ± 0.033	**0.006**	**0.011**	0.085
***SFA* ^a^**	**32.95 ± 0.772**	**34.00 ± 0.734**	**32.67 ± 0.511**	**34.28 ± 0.980**	0.341	0.764	0.795
C16:1 *n-9*	4.78 ± 0.195	3.55 ± 0.223	4.24 ± 0.381	4.09 ± 0.310	**<0.001**	**0.009**	**0.023**
C18:1 *n-7*	3.87 ± 0.452	3.19 ± 0.328	3.53 ± 0.242	3.53 ± 0.548	0.154	0.278	0.208
C18:1 *n-9*	34.74 ± 0.443	33.72 ± 0.792	33.11 ± 0.269	35.35 ± 0.716	0.264	0.737	0.600
C20:1 *n-9*	0.82 ± 0.052	1.12 ± 0.039	0.95 ± 0.107	0.99 ± 0.060	**<0.001**	**0.006**	**0.003**
C22:1 *n-9*	0.10 ± 0.040	0.22 ± 0.016	0.15 ± 0.044	0.17 ± 0.038	**0.006**	**0.003**	**0.035**
***MUFA* ^b^**	**44.31 ± 0.663**	**41.79 ± 1.039**	**41.97 ± 0.466**	**44.13 ± 1.264**	0.080	0.708	0.962
C18:2 *n-6*	12.56 ± 0.421	14.57 ± 0.641	14.28 ± 0.620	12.85 ± 0.627	**0.009**	0.987	0.849
C18:3 *n-3*	0.91 ± 0.063	0.87 ± 0.040	0.90 ± 0.056	0.88 ± 0.050	0.230	0.097	0.922
C18:3 *n-6*	0.03 ± 0.011	0.07 ± 0.014	0.06 ± 0.014	0.04 ± 0.018	**0.013**	0.041	0.958
C20:2	0.04 ± 0.007	0.05 ± 0.006	0.04 ± 0.008	0.04 ± 0.006	0.187	0.941	0.17
C20:3 *n-3*	0.27 ± 0.022	0.19 ± 0.008	0.25 ± 0.028	0.21 ± 0.013	0.001	0.948	0.083
C20:3 *n-6*	0.37 ± 0.013	0.37 ± 0.008	0.37 ± 0.013	0.37 ± 0.009	0.867	0.351	0.648
C20:4 *n-6*	2.46 ± 0.234	1.44 ± 0.052	2.15 ± 0.347	1.75 ± 0.168	**<0.001**	0.971	**0.007**
C20:5 *n-3*	0.33 ± 0.063	0.22 ± 0.012	0.31 ± 0.061	0.24 ± 0.030	**0.044**	0.309	0.330
C22:4 *n-6*	0.50 ± 0.052	0.32 ± 0.012	0.44 ± 0.066	0.37 ± 0.034	**<0.001**	0.251	**0.032**
C22:5 *n-3*	0.12 ± 0.012	0.10 ± 0.004	0.11 ± 0.009	0.11 ± 0.010	0.065	0.098	0.228
C22:5 *n-6*	0.14 ± 0.018	0.09 ± 0.006	0.12 ± 0.021	0.10 ± 0.014	**0.022**	**0.028**	0.209
C22:6 *n-3*	0.11 ± 0.027	0.17 ± 0.099	0.13 ± 0.023	0.14 ± 0.026	0.159	0.080	0.889
***PUFA* ^c^**	**17.82 ± 0.773**	**18.49 ± 0.686**	**19.19 ± 0.411**	**17.12 ± 0.572**	0.450	0.589	0.153
**Fatty Acid Indices**							
∑n−6	16.04 ± 0.701	16.87 ± 0.659	17.42 ± 0.434	15.48 ± 0.527	0.285	0.980	0.258
∑n−3	1.74 ± 0.174	1.55 ± 0.033	1.71 ± 0.151	1.58 ± 0.097	0.076	0.094	0.544
PUFA/SFA	0.54 ± 0.029	0.55 ± 0.023	0.59 ± 0.018	0.50 ± 0.017	0.541	0.589	0.486
n-6/n-3	9.59 ± 0.976	10.86 ± 0.272	10.54 ± 0.931	9.91 ± 0.529	0.161	0.140	0.936

^A^ animals with age ranging from 13–24 months; ^B^ animals with age older than 24 months; ^a^ Saturated fatty acids; ^b^ Monounsaturated fatty acids; ^c^ Polyunsaturated fatty acids; ^e^ animal age was estimated based on tooth eruption, replacement, and wear patterns; SEM: standard error of the mean.

## Data Availability

Data sharing is not applicable to this article.

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
