# Peer review of "Meat Fatty Acid Composition of Wild Boars Hunted in Romania in Relationship to Gender and Age-Class"

_animals, 2022, doi:10.3390/ani12070810_

Round 1

Reviewer 1 Report

The paper presents interesting results concerning the carcass composition of wild boars , the proximate analysis and fatty acid profile of their meat as affected by the age and gender. The Abstract is concise yet revealing the main findings of the study. The Introduction provides the necessary and sufficient information about the need to conduct the research. The aim of the study however in not clearly defined and should be added at the end of the Introduction. A specific remark here: Line 77-79, please rephrase the statement as: Although numerous studies on the fatty acid composition content of wild boar meat have been conducted [35], these traits are still unknown for the wild boars hunted in Romania and serving  as raw material for industrial processors.

The experimental design and the description of the material and methods should be improved. The authors present results about the carcass of the boars ( weigh, carcass weight, dressing percentage), however nowhere in the Material and method section this is described. They need to provide information how they took these measurements, how the dressing percentage is calculated, etc. A specific remark here and for all the remaining text: Please use either proximate composition, or chemical composition, not proximate chemical.

The statistical evaluation is rather not correct. The authors state that the carcass composition traits were evaluated through one way ANOVA. This is rather t-test applied for both age and gender. In my opinion the authors should have applied two-way ANOVA as they did for the proximate and fatty acid composition and analyse the effect of gender, age and their interaction.

The two way ANOVA applied for the proximate and fatty acid analysis should be presented in another way. The probability ( significance levels) should be presented for each of the factors and their interaction as done in the table in the Effects column. There is no need to present probability values for the factors separately as you done for the male and female , subadults and adults. Also there is no need of multiple comparisons because each factors has two levels only. Usually when performing ANOVA, standard deviations are not presented. Only RMSE or the standard error of the model is reported.

For the probability you do not need to write asterisks or NS, because the value of the probability itself shows is there is significance or not and on what level.

Figure 1 is not needed. The results that it presents are already presented in TAble 4. Avoid presenting the same data in  both figures and tables.

The results are otherwise well presented and well discussed. The conclusions are derived form the results.

Author Response

Dear Reviewer,

We thank you for reviewing our manuscript and for your suggestions made to improve it. Please see below the answers (in red) to your comments.

  1. The aim of the study however in not clearly defined and should be added at the end of the Introduction.

The objectives of this research was rephrased in order to be clearly defined and was added at the end of the Introduction.

Plants (grasses, leaves, roots, bushes, seeds, forest fruits) and, less frequently, bird eggs, snails, insects, earthworms, larvae, and beetles are the primary food sources for wild boars in their natural habitats [31, 38]. Due to the natural habitat in which wild animals thrive, venison is frequently referred to as an organic food. In light of this information, the objectives of this research were to investigate the carcass characteristics, chemical composition, and fatty acid profile of wild boar Biceps femoris muscle and, also to analyse the effects of gender, age-class, and their interaction on the mentioned traits.

  1. A specific remark here: Line 77-79, please rephrase the statement as: Although numerous studies on the fatty acid composition content of wild boar meat have been conducted [35], these traits are still unknown for the wild boars hunted in Romania and serving as raw material for industrial processors.

The sentence was rephrased: 

Despite the fact that numerous studies on the fatty acid composition of wild boar meat have been conducted in various parts of the world [36, 39, 45, 46], no studies on wild boar meat hunted in Romania and used as a raw material for industrial processors have been done.

  1. The experimental design and the description of the material and methods should be improved. The authors present results about the carcass of the boars ( weigh, carcass weight, dressing percentage), however nowhere in the Material and method section this is described. They need to provide information how they took these measurements, how the dressing percentage is calculated, etc. A specific remark here and for all the remaining text: Please use either proximate composition, or chemical composition, not proximate chemical.

The experimental design and the description of the material and methods was improved. The following paragraph was added:

The carcass parameters for all animals were determined using a calibrated scale with a limit of 150 kg, as follows: Warm carcass weight represents the weight of animals shot without organs but with kidneys 5 hours after shooting; cold carcass weight represents the weight of animals shot without organs but with kidneys 24 hours after slaughtering; and dressing percentage represents the cold carcass as a percentage of its undressed weight. The animal's age was calculated using tooth eruption, which takes over 26 months to complete, replacement, and wear patterns [33, 2].

4.The statistical evaluation is rather not correct. The authors state that the carcass composition traits were evaluated through one way ANOVA. This is rather t-test applied for both age and gender. In my opinion the authors should have applied two-way ANOVA as they did for the proximate and fatty acid composition and analyse the effect of gender, age and their interaction. The two way ANOVA applied for the proximate and fatty acid analysis should be presented in another way. The probability (significance levels) should be presented for each of the factors and their interaction as done in the table in the Effects column. There is no need to present probability values for the factors separately as you done for the male and female , subadults and adults. Also there is no need of multiple comparisons because each factors has two levels only. Usually when performing ANOVA, standard deviations are not presented. Only RMSE or the standard error of the model is reported.

The statistical analysis has been redone according to your proposal (two-way ANOVA), and the new table were insert in the article (Table 2, 3 and 4).

Table 2. Wild boar’s carcass distribution (mean ±SEM).

Trait

Gender

Age-classe

Effect

Male

(n = 27)

Female

(n = 49)

SubadultsA

(n = 39)

AdultsB

(n = 37)

gender

age-class

gender x

age-class

Undressed weighta (kg)

92.77 ± 9.36

76.76 ± 4.19

71.89 ± 3.77

97.64 ± 7.10

0.03

0.05

0.18

Warm carcass weightb (kg)

74.90 ± 8.29

59.99 ± 3.51

56.49 ± 3.87

78.40 ± 6.35

0.03

0.03

0.13

Cold carcass weightc (kg)

72.01 ± 8.10

57.70 ± 3.52

53.96 ± 3.69

75.75 ± 6.13

0.03

0.04

0.16

Dressing percentaged (%)

77.09 ± 1.33

75.05 ± 0.57

74.79 ± 1.13

77.36 ± 0.75

0.16

0.88

0.73

  • a weight of the freshly shot animal.
  • b without organs but with kidneys, 5 hours after shooting
  • c without organs but with kidneys, 24 hours after slaughtering
  • d cold carcass as a proportion of undressed weight
  • e animal age was estimated based on tooth eruption, replacement and wear patterns [32];
  • A animals with age ranging from 13 – 24 months;
  • B animals with age older than 24 months;
  • SEM = standard error of the mean; ns = P > 0.05; significant = P ≤ 0.05;

Table 3. Chemical composition (mean ± SEM) of Biceps femoris muscle from wild boars hunted in Romania.

Traits

Gender

Age-classe

Effect

Male

(n = 27)

Female

(n = 49)

SubadultsA

(n = 39)

AdultsB

(n = 37)

gender

age-class

gender x

age-class

Moisture (%)

64.59 ± 2.12

61.83 ± 1.00

61.86 ± 1.75

64.55 ± 1.58

0.262

0.274

0.315

Protein (%)

22.76 ± 0.56

22.00 ± 0.53

22.78 ± 0.59

21.99 ± 0.50

0.365

0.347

0.494

Fat (%)

5.56 ± 0.75

6.55 ± 0.82

4.52 ± 0.23

7.60 ± 0.58

0.129

0.001

0.525

Ash (%)

1.14 ± 0.05

1.06 ± 0.17

0.99 ± 0.11

1.20 ± 0.13

0.676

0.243

0.188

A animal with age ranging from 13 – 24 months;

B animals with age older than 24 months;

e animal age was estimated based on tooth eruption, replacement and wear patterns [32];

SEM = standard error of the mean; ns = P > 0.05; significant = P ≤ 0.001;

Table 4. Effect of gender and age-class on the fatty acid composition (% of total fatty acids, mean ± SEM) of M. Biceps femoris, corrected for carcass weight and muscle fat content component.

Gender

.

Age-classe

.

Effect

Male

(n = 27)

Female

(n = 49)

SubadultsA

(n = 39)

AdultsB

(n = 37)

gender

age-class

gender x

age-class

C14:0

0.24 ± 0.161

0.31 ± 0.303

0.27 ± 0.105

0.28 ± 0.128

0.197

0.017

0.964

C15:0

0.01 ± 0.007

0.04 ± 0.010

0.03 ± 0.011

0.02 ± 0.013

0.033

0.987

0.864

C16:0

21.44 ± 0.501

21.62 ± 0.630

20.93 ± 0.215

22.13 ± 0.736

0.745

0.976

0.592

C17:0

0.23 ± 0.010

0.24 ± 0.009

0.23 ± 0.012

0.23 ± 0.008

0.605

0.298

0.229

C18:0

10.89 ± 0.192

11.50 ± 0.168

10.99 ± 0.284

11.39 ± 0.190

0.065

0.761

0.643

C20:0

0.02 ± 0.008

0.06 ± 0.017

0.04 ± 0.010

0.04 ± 0.020

0.052

0.278

0.404

C22:0

0.12 ± 0.029

0.23 ± 0.020

0.18 ± 0.040

0.18 ± 0.033

0.006

0.011

0.085

SFAa

32.95 ± 0.772

34.00 ± 0.734

32.67 ± 0.511

34.28 ± 0.980

0.341

0.764

0.795

C16:1 n-9

4.78 ± 0.195

3.55 ± 0.223

4.24 ± 0.381

4.09 ± 0.310

< 0.001

0.009

0.023

C18:1 n-7

3.87 ± 0.452

3.19 ± 0.328

3.53 ± 0.242

3.53 ± 0.548

0.154

0.278

0.208

C18:1 n-9

34.74 ± 0.443

33.72 ± 0.792

33.11 ± 0.269

35.35 ± 0.716

0.264

0.737

0.600

C20:1 n-9

0.82 ± 0.052

1.12 ± 0.039

0.95 ± 0.107

0.99 ± 0.060

< 0.001

0.006

0.003

C22:1 n-9

0.10 ± 0.040

0.22 ± 0.016

0.15 ± 0.044

0.17 ± 0.038

0.006

0.003

0.035

MUFAb

44.31 ± 0.663

41.79 ± 1.039

41.97 ± 0.466

44.13 ± 1.264

0.080

0.708

0.962

C18:2 n-6

12.56 ± 0.421

14.57 ± 0.641

14.28 ± 0.620

12.85 ± 0.627

0.009

0.987

0.849

C18:3 n-3

0.91 ± 0.063

0.87 ± 0.040

0.90 ± 0.056

0.88 ± 0.050

0.230

0.097

0.922

C18:3 n-6

0.03 ± 0.011

0.07 ± 0.014

0.06 ± 0.014

0.04 ± 0.018

0.013

0.041

0.958

C20:2

0.04 ± 0.007

0.05 ± 0.006

0.04 ± 0.008

0.04 ± 0.006

0.187

0.941

0.17

C20:3 n-3

0.27 ± 0.022

0.19 ± 0.008

0.25 ± 0.028

0.21 ± 0.013

0.001

0.948

0.083

C20:3 n-6

0.37 ± 0.013

0.37 ± 0.008

0.37 ± 0.013

0.37 ± 0.009

0.867

0.351

0.648

C20:4 n-6

2.46 ± 0.234

1.44 ± 0.052

2.15 ± 0.347

1.75 ± 0.168

< 0.001

0.971

0.007

C20:5 n-3

0.33 ± 0.063

0.22 ± 0.012

0.31 ± 0.061

0.24 ± 0.030

0.044

0.309

0.330

C22:4 n-6

0.50 ± 0.052

0.32 ± 0.012

0.44 ± 0.066

0.37 ± 0.034

< 0.001

0.251

0.032

C22:5 n-3

0.12 ± 0.012

0.10 ± 0.004

0.11 ± 0.009

0.11 ± 0.010

0.065

0.098

0.228

C22:5 n-6

0.14 ± 0.018

0.09 ± 0.006

0.12 ± 0.021

0.10 ± 0.014

0.022

0.028

0.209

C22:6 n-3

0.11 ± 0.027

0.17 ± 0.099

0.13 ± 0.023

0.14 ± 0.026

0.159

0.080

0.889

PUFAc

17.82 ± 0.773

18.49 ± 0.686

19.19 ± 0.411

17.12 ± 0.572

0.450

0.589

0.153

Fatty Acid Indices

16.04 ± 0.701

16.87 ± 0.659

17.42 ± 0.434

15.48 ± 0.527

0.285

0.980

0.258

1.74 ± 0.174

1.55 ± 0.033

1.71 ± 0.151

1.58 ± 0.097

0.076

0.094

0.544

PUFA/SFA

0.54 ± 0.029

0.55 ± 0.023

0.59 ± 0.018

0.50 ± 0.017

0.541

0.589

0.486

n-6/n-3

9.59 ± 0.976

10.86 ± 0.272

10.54 ± 0.931

9.91 ± 0.529

0.161

0.140

0.936

A animal’s with age ranging from 13 – 24 months; B animals with age older than 24 months;

a Saturated fatty acid; b Monounsaturated fatty acids; c Polyunsaturated fatty acids;

e animal age was estimated based on tooth eruption, replacement and wear patter [32];

SEM = standard error of the mean.

For the probability you do not need to write asterisks or NS, because the value of the probability itself shows is there is significance or not and on what level.

Done.

Figure 1 is not needed. The results that it presents are already presented in TAble 4. Avoid presenting the same data in  both figures and tables.

Figure 1 was eliminated in order to avoid presenting the same data in both figures and tables. The results from figure 1 are presented in Table 4.

Once again, we appreciate your time on reviewing this manuscript. Hopefully, the revised version of this manuscript is acceptable for this special issue of Animals.

Kind regards,

Dr. Florin LIPSA

Reviewer 2 Report

I have gone through the manuscript and found it interesting. The authors studied the most appealing area of meat sciences, which should be explored. The objective of authors was to explore the meat quality parameters of game meat of wild boar over seven seasons that are novel and innovative.  However, there are a lot of flaws in experimental design, grammar, results presentation, and discussion. Authors need to address all the issues before publication

Simple summary

Line 18-20: Remove the sentence ‘Meat is considered an essential component of a balanced human diet. Due to the rising trend for healthy products, consumers tend to search for healthier alternatives hence the growth in popularity of game meat’

Line 22: replace domestic with domesticated

Line 22-24: the details of experimental unit is missing ‘The present study aimed to assess the carcass characteristics and chemical composition, and also to investigate the effect of gender and age on the intramuscular fatty acid profile’

Line 24-25: sentence is not clear  ‘The age-class factor influenced the carcass traits and the yield, adults were heavier, having a favourable yield compared to the younger specimens’

Line 25: what do you mean by ‘The differences induced by….’ Is it about carcass parameters and yield? If yes please mention

Line 26: ‘For chemical composition…’ chemical composition of what ?

Line 26-28: these sentences are so confusing, please rewrite ‘For chemical composition, the percentage of protein, ash and water did not present significant differences influenced by gender and age, only the fat content was influenced by these two factors, adults and females having a greater content than subadults and males’

Line 30: replace ‘highlighted’ with ‘present in meat’

Abstract

Experimental treatments are not clear

Objectives are not clear

Sample collection is not present

Line 32: ‘Biceps femoris muscle’ from right side or left side of carcasses?

Line 34: replace ‘seven seasons, only in winter’ with ‘ seven winter seasons consecutively

Line 34-36: replace ‘The postmortem undressed carcass weight of analyzed animals was affected by age-class (P ≤ 0.05), this influence being highlighted and for warm and cold carcass weight (P ≤ 0.05)’ with ‘The postmortem undressed warm and cold carcass weight of was affected by age of the animals (P ≤ 0.05)’

Line 36: what do you mean by extractable fat?

Line 36-37: ‘……….extractable fat content was higher ……..’ was higher in ?

Line 38: what is IMF? Do you mean intermuscular fat? If yes, please give full term on first seen

Line 39: use full term of MUFA and PUFA on first seen

Line 38-39: give specific results instead of this general statement ‘For IMF content, gender, age class and their interaction appear to influence the certain fatty acid composition, MUFA and PUFA fatty acids…’

Line 41: what do you mean by ‘P / S acids’

Introduction

Introduction is poorly written, without focusing the objective.

Line 46: replace ‘and, with it, of the increased risk of diet-related diseases of this century’ with ‘ and increased diseases risk in conventional farm meat’

Line 47-50:  Sentence too long to understand ‘For these reasons there has been an increase in the popularity of game meat; although traditionally the consumption of game meat is higher in mountainous areas, specifically for hunting, as the animals harvested from hunting parties are consumed by hunters and their families, as well as those who have a certain degree of knowledge and are familiar with the products [9, 17]’ please rephrase it

Line 52: replace ‘quantities’ with ‘ consumption’

Line 52: remove ‘continent’ from sentence

Line 53: remove ‘the main influencers being age and sex’

Line 54: ‘meat’ you mean game meat? If yes include it

Line 54: replace ‘as a result of their preferences for’ with ‘due to’

Line 60-61: rephrase it ‘and the quality of it, especially from the point of health-related fat quality meat indices’

Line 67-70: sentences are confusing ‘Of high importance in the quality of game meat are the fat content, the fatty acid profile, also the wholesome protein and due to this quality parameters this type of meat is considered to be healthier compared to meat from domestic animals’ rewrite

Line 69-70: domesticated not domestic

Line 71: start sentence from ‘Intramuscular fat is well known…’

Line 72-73: remove sentence ‘wild boar meat being described with an intense, sweet and nutty flavour when having the balanced chemical components’

Line 77: merge this paragraph with previous one ‘Although numerous studies on the fatty acid composition…..’

Line 77: you referred only one study but in sentence you are claiming ‘numerous studies’. Please correct it

Line 77-79: rewrite sentence ‘Although numerous studies on the fatty acid composition content of wild boar meat have been conducted [35] wild boars hunted in Romania and served as raw material for industrial processors it is unknown these traits’ the new sentence could be ‘‘Although numerous studies on the fatty acid composition contents of wild boar meat have been conducted [35] but studies on wild boars meat hunted in Romania and served as raw material for industrial processors are not yet conducted’

Line 80-82: rewrite objective

Line 83-89: useless information

Materials and Methods

Need to improve sentence structure and grammar. Authors failed to explain the experimental treatments, for example how animals were classified on the basis of age? Number of animals in each treatment is different, authors needs to arrange and analyses the data by keeping the same number of animals in each treatment. Furthermore, authors need to address factorial design analysis model in statistical analysis portion

Line 99: remove ‘forms’ after concentrate

Line 104: animals were shoted not Animals were shot

Line 109: use italic ‘m. Biceps femoris’ throughout study

Line 124: remove GLC

Line 127: check reference Folch et al. [833

Line 127: repetition ‘fatty acid methyl esters (FAME)’ on line 129

Line 130: remove (FID)

Results

It's not only results, it contains both results and discussion. Please correct it accordingly

The English language is poor. Please contact some professionals to improve. Discussion should be with proper reasoning and citations. Results are ok but need to present in a better way. Please correct the whole results and discussion as suggested below

Line 152-154: results are confusing. At first seen, it seems that results are significant. Please correct it and rewrite the sentences ‘Wild boar’s undressed weight was greater at males 152 than females, the average values being 75 vs. 60 kg, with a favourable yield in the case of 153 the males; however, these differences were not statistically significant (P > 0.05)’

Line 152-160: I can't find justification of results. Please justify results with proper reasons and references.

Line 172-75: contradictory statements ‘The percentage of protein, ash and water was not influenced by gender and age class (P > 0.05). With the increasing age of the animals, the total lipid content increased significantly (P ≤ 0.001), meat from adults being fatter than meat from subadult wild boars’ please correct it

Line 175-184: very poor justification. Please correct it accordingly

Conclusion

Although conclusion is drawn from results but poor presentation of results unable me to understand the exact meaning. Rewrite the conclusion again by keeping in view the grammar, sentence structure and technical presentation.

Author Response

Dear Reviewer,

We thank you for reviewing our manuscript and for your suggestions made to improve it. Please see below the answers (in red) to your comments.

Simple summary

Line 18-20: Remove the sentence ‘Meat is considered an essential component of a balanced human diet. Due to the rising trend for healthy products, consumers tend to search for healthier alternatives hence the growth in popularity of game meat’

The sentence was removed.

Line 22: replace domestic with domesticated

The term domestic has been replaced in the article with domesticated.

Line 22-24: the details of experimental unit is missing ‘The present study aimed to assess the carcass characteristics and chemical composition, and also to investigate the effect of gender and age on the intramuscular fatty acid profile’

The paragraph was completed and rephrased: The present study aimed to assess the carcass characteristics and chemical composition, and also to investigate the effect of gender and age-class on the intramuscular fatty acid profile of wild boar meat harvested during the 2010-2017 winter hunting seasons in the Frasin District hunting area (Romania).  

Line 24-25: sentence is not clear  ‘The age-class factor influenced the carcass traits and the yield, adults were heavier, having a favourable yield compared to the younger specimens’

The sentence was rephrased: The age-class factor influenced the carcass traits (P≤0.05), but not the yield, although overall, adults were heavier and have a better yield compared to the younger specimens (77.36 vs. 74.79 kg).

Line 25: what do you mean by ‘The differences induced by….’ Is it about carcass parameters and yield? If yes please mention

The requested completions have been made: The differences induced by gender for carcass traits and dressing yield were present but not statistically significant (P>0.05).

Line 26: ‘For chemical composition…’ chemical composition of what?

For the chemical composition of wild boar meat….

Line 26-28: these sentences are so confusing, please rewrite ‘For chemical composition, the percentage of protein, ash and water did not present significant differences influenced by gender and age, only the fat content was influenced by these two factors, adults and females having a greater content than subadults and males’

The paragraph was rephrased: For the chemical composition of wild boar meat, the percentages of protein, fat, ash and water did not present significant differences (P>0.05) within the factors applied (gender, age-class). Regarding the studied effects, results showed that the wild boars’ age-class influenced significantly (P≤0.001) the amount of meat total fat, with the adults having the highest fat content than subadults (7.60% vs. 4.52%).

Line 30: replace ‘highlighted’ with ‘present in meat’

Corrected. As for the fatty acid profile, a high ratio of PUFA/SFA (P/S – polyunsaturated fatty acids/saturated fatty acids) and n-6/n-3 PUFA was present in the meat, showing good functional properties for human health.

Abstract

Experimental treatments are not clear. Objectives are not clear. Sample collection is not present

The following paragraphs have been added to address the 3 comments raised by the reviewer:

The present study evaluates the influence of gender and age-class on the carcass traits, meat chemical composition, and fatty acid profiles of randomly hunt-harvested wild boars from Romania's Frasin District hunting area, which is an important part of the stock density management strategy. Only 76 wild boars were chosen to participate in the experimental design from a total of 94 wild boars hunted over seven winter seasons consecutively, based on two main criteria: carcass right part intact and animal health status (suitable for human consumption) as determined by the veterinary health notice issued by the Food Safety Authority, with all wild boars being evaluated before slaughtering. After sanitary inspection and within the first 24 hours post-mortem, meat samples were taken in duplicate from the right side of wild boar carcasses (m. Biceps femoris).

The aim of this study was to compare the carcass characteristics, chemical composition, and fatty acid profile of wild boar Biceps femoris muscle obtained from the right side of carcasses, as well as the effects of gender, age-class, and their interaction on these traits, in a context where venison is frequently regarded as organic food by consumers due to the natural habitat in which wild animals live.

Line 32: ‘Biceps femoris muscle’ from right side or left side of carcasses?

The sentence was completed with the required information: Biceps femoris muscle obtained from the right side of carcasses,….

Line 34: replace ‘seven seasons, only in winter’ with ‘ seven winter seasons consecutively

Corrected: Only 76 wild boars were chosen to participate in the experimental design from a total of 94 wild boars hunted over seven winter seasons consecutively,….

Line 34-36: replace ‘The postmortem undressed carcass weight of analyzed animals was affected by age-class (P ≤ 0.05), this influence being highlighted and for warm and cold carcass weight (P ≤ 0.05)’ with ‘The postmortem undressed warm and cold carcass weight of was affected by age of the animals (P ≤ 0.05)’

The sentence was rephrased: The age-class of the animals had an effect on the postmortem undressed warm and cold carcass weights (P 0.05).

Line 36: what do you mean by extractable fat?

Clarification: extractable fat is intramuscular fat. I have changed in the article extractable fat with intramuscular fat.

Line 36-37: ‘……….extractable fat content was higher ……..’ was higher in ?

The sentence was completed: Considering the proximal chemical composition, only the intramuscular fat (IMF) content was higher in adult meat samples compared to young animals.

Line 38: what is IMF? Do you mean intermuscular fat? If yes, please give full term on first seen

Corrected: ….only the intramuscular fat (IMF) …

Line 39: use full term of MUFA and PUFA on first seen

Resolved. Full term of MUFA and PUFA are introduced in Simple summary.

Line 38-39: give specific results instead of this general statement ‘For IMF content, gender, age class and their interaction appear to influence the certain fatty acid composition, MUFA and PUFA fatty acids…’

Corrected: For IMF content, gender, age class and their interaction influenced mainly MUFA fatty acids, in particular C16:1 n-9, C20:1 n-9 and C22:1 n-9, and also PUFA fatty acids C18:3 n-6, C20:3 n-3, C20:4 n-6, C20:5 n-3, C22:4 n-6 and C22:5 n-6.

Line 41: what do you mean by ‘

Full term of P / S acids’ is introduced in Simple summary: …PUFA/SFA (P/S – polyunsaturated fatty acids/saturated fatty acids)…

Introduction

Introduction is poorly written, without focusing the objective.

Introduction was improved and the objectives are clearly presented at the end of Introduction.

Line 46: replace ‘and, with it, of the increased risk of diet-related diseases of this century’ with ‘ and increased diseases risk in conventional farm meat’

The sentence was rephrased: Consumers today are encouraged to look for alternative meat types due to the changes in their eating habits and the increased disease risk associated with conventional farm meat [35, 44].

Line 47-50:  Sentence too long to understand ‘For these reasons there has been an increase in the popularity of game meat; although traditionally the consumption of game meat is higher in mountainous areas, specifically for hunting, as the animals harvested from hunting parties are consumed by hunters and their families, as well as those who have a certain degree of knowledge and are familiar with the products [9, 17]’ please rephrase it

The sentence was rephrased: For these reasons, game meat has grown in popularity, despite the fact that consumption of game meat has traditionally been higher in mountainous areas. There is a link between these areas and hunting activity, as harvested animals are consumed by hunters and their families, as well as those who have a certain degree of knowledge and are familiar with game meat products [8, 18].

Line 52: replace ‘quantities’ with ‘ consumption’

Corrected.

Line 52: remove ‘continent’ from sentence

Removed.

Line 53: remove ‘the main influencers being age and sex’

Deleted.

Line 54: ‘meat’ you mean game meat? If yes include it

Corrected.

Line 54: replace ‘as a result of their preferences for’ with ‘due to’

Replaced.

Line 60-61: rephrase it ‘and the quality of it, especially from the point of health-related fat quality meat indices’

The sentence was rephrased: ….and its quality, particularly in terms of health-related fat quality meat indicators [19].

Line 67-70: sentences are confusing ‘Of high importance in the quality of game meat are the fat content, the fatty acid profile, also the wholesome protein and due to this quality parameters this type of meat is considered to be healthier compared to meat from domestic animals’ rewrite

Line 69-70: domesticated not domestic

Corrected.

Line 71: start sentence from ‘Intramuscular fat is well known…’

Done.

Line 72-73: remove sentence ‘wild boar meat being described with an intense, sweet and nutty flavour when having the balanced chemical components’

Sentence deleted.

Line 77: merge this paragraph with previous one ‘Although numerous studies on the fatty acid composition…..’

The sentence was rephrased: Despite the fact that numerous studies on the fatty acid composition of wild boar meat have been conducted in various parts of the world [36, 39, 45, 46], no studies on wild boar meat hunted in Romania and used as a raw material for industrial processors have been done.

Line 77: you referred only one study but in sentence you are claiming ‘numerous studies’. Please correct it

Reference added .. [36, 39, 45, 46]

Line 77-79: rewrite sentence ‘Although numerous studies on the fatty acid composition content of wild boar meat have been conducted [35] wild boars hunted in Romania and served as raw material for industrial processors it is unknown these traits’ the new sentence could be ‘‘Although numerous studies on the fatty acid composition contents of wild boar meat have been conducted [35] but studies on wild boars meat hunted in Romania and served as raw material for industrial processors are not yet conducted’

The sentence was rephrased: Despite the fact that numerous studies on the fatty acid composition of wild boar meat have been conducted in various parts of the world [36, 39, 45, 46], no studies on wild boar meat hunted in Romania and used as a raw material for industrial processors have been done.

Line 80-82: rewrite objective

Objectives were rewritten

Line 83-89: useless information

Deleted.

Materials and Methods

Need to improve sentence structure and grammar.

Sentence structure and grammar was enhanced by an English proof-reader.

Authors failed to explain the experimental treatments, for example how animals were classified on the basis of age? Number of animals in each treatment is different, authors needs to arrange and analyses the data by keeping the same number of animals in each treatment. Furthermore, authors need to address factorial design analysis model in statistical analysis portion.

The experimental design and the description of the material and methods was improved. The following paragraphs were added:

The study included 76 wild boars (Sus scrofa ferus), which were harvested under Romania's national game, hunting, and wildlife protection legislation [51] during the 2010–2017 hunting seasons (winter), with the purpose of stock density control in the Frasin District hunting area. At the beginning of this study, the livestock spring population of wild boars had an estimated value of 95 animals (Table 1).

The carcass parameters for all animals were determined using a calibrated scale with a limit of 150 kg, as follows: Warm carcass weight represents the weight of animals shot without organs but with kidneys 5 hours after shooting; cold carcass weight represents the weight of animals shot without organs but with kidneys 24 hours after slaughtering; and dressing percentage represents the cold carcass as a percentage of its undressed weight. The animal's age was calculated using tooth eruption, which takes over 26 months to complete, replacement, and wear patterns [33, 2].

The statistical analysis has been redone using two-way ANOVA, and the new tables replace the older ones in the article (Tables 2, 3 and 4).

Table 2. Wild boar’s carcass distribution (mean ±SEM).

Trait

Gender

Age-classe

Effect

Male

(n = 27)

Female

(n = 49)

SubadultsA

(n = 39)

AdultsB

(n = 37)

gender

age-class

gender x

age-class

Undressed weighta (kg)

92.77 ± 9.36

76.76 ± 4.19

71.89 ± 3.77

97.64 ± 7.10

0.03

0.05

0.18

Warm carcass weightb (kg)

74.90 ± 8.29

59.99 ± 3.51

56.49 ± 3.87

78.40 ± 6.35

0.03

0.03

0.13

Cold carcass weightc (kg)

72.01 ± 8.10

57.70 ± 3.52

53.96 ± 3.69

75.75 ± 6.13

0.03

0.04

0.16

Dressing percentaged (%)

77.09 ± 1.33

75.05 ± 0.57

74.79 ± 1.13

77.36 ± 0.75

0.16

0.88

0.73

  • a weight of the freshly shot animal.
  • b without organs but with kidneys, 5 hours after shooting
  • c without organs but with kidneys, 24 hours after slaughtering
  • d cold carcass as a proportion of undressed weight
  • e animal age was estimated based on tooth eruption, replacement and wear patterns [32];
  • A animals with age ranging from 13 – 24 months;
  • B animals with age older than 24 months;
  • SEM = standard error of the mean; ns = P > 0.05; significant = P ≤ 0.05;

Table 3. Chemical composition (mean ± SEM) of Biceps femoris muscle from wild boars hunted in Romania.

Traits

Gender

Age-classe

Effect

Male

(n = 27)

Female

(n = 49)

SubadultsA

(n = 39)

AdultsB

(n = 37)

gender

age-class

gender x

age-class

Moisture (%)

64.59 ± 2.12

61.83 ± 1.00

61.86 ± 1.75

64.55 ± 1.58

0.262

0.274

0.315

Protein (%)

22.76 ± 0.56

22.00 ± 0.53

22.78 ± 0.59

21.99 ± 0.50

0.365

0.347

0.494

Fat (%)

5.56 ± 0.75

6.55 ± 0.82

4.52 ± 0.23

7.60 ± 0.58

0.129

0.001

0.525

Ash (%)

1.14 ± 0.05

1.06 ± 0.17

0.99 ± 0.11

1.20 ± 0.13

0.676

0.243

0.188

A animal with age ranging from 13 – 24 months;

B animals with age older than 24 months;

e animal age was estimated based on tooth eruption, replacement and wear patterns [32];

SEM = standard error of the mean; ns = P > 0.05; significant = P ≤ 0.001;

Table 4. Effect of gender and age-class on the fatty acid composition (% of total fatty acids, mean ± SEM) of M. Biceps femoris, corrected for carcass weight and muscle fat content component.

Gender

.

Age-classe

.

Effect

Male

(n = 27)

Female

(n = 49)

SubadultsA

(n = 39)

AdultsB

(n = 37)

gender

age-class

gender x

age-class

C14:0

0.24 ± 0.161

0.31 ± 0.303

0.27 ± 0.105

0.28 ± 0.128

0.197

0.017

0.964

C15:0

0.01 ± 0.007

0.04 ± 0.010

0.03 ± 0.011

0.02 ± 0.013

0.033

0.987

0.864

C16:0

21.44 ± 0.501

21.62 ± 0.630

20.93 ± 0.215

22.13 ± 0.736

0.745

0.976

0.592

C17:0

0.23 ± 0.010

0.24 ± 0.009

0.23 ± 0.012

0.23 ± 0.008

0.605

0.298

0.229

C18:0

10.89 ± 0.192

11.50 ± 0.168

10.99 ± 0.284

11.39 ± 0.190

0.065

0.761

0.643

C20:0

0.02 ± 0.008

0.06 ± 0.017

0.04 ± 0.010

0.04 ± 0.020

0.052

0.278

0.404

C22:0

0.12 ± 0.029

0.23 ± 0.020

0.18 ± 0.040

0.18 ± 0.033

0.006

0.011

0.085

SFAa

32.95 ± 0.772

34.00 ± 0.734

32.67 ± 0.511

34.28 ± 0.980

0.341

0.764

0.795

C16:1 n-9

4.78 ± 0.195

3.55 ± 0.223

4.24 ± 0.381

4.09 ± 0.310

< 0.001

0.009

0.023

C18:1 n-7

3.87 ± 0.452

3.19 ± 0.328

3.53 ± 0.242

3.53 ± 0.548

0.154

0.278

0.208

C18:1 n-9

34.74 ± 0.443

33.72 ± 0.792

33.11 ± 0.269

35.35 ± 0.716

0.264

0.737

0.600

C20:1 n-9

0.82 ± 0.052

1.12 ± 0.039

0.95 ± 0.107

0.99 ± 0.060

< 0.001

0.006

0.003

C22:1 n-9

0.10 ± 0.040

0.22 ± 0.016

0.15 ± 0.044

0.17 ± 0.038

0.006

0.003

0.035

MUFAb

44.31 ± 0.663

41.79 ± 1.039

41.97 ± 0.466

44.13 ± 1.264

0.080

0.708

0.962

C18:2 n-6

12.56 ± 0.421

14.57 ± 0.641

14.28 ± 0.620

12.85 ± 0.627

0.009

0.987

0.849

C18:3 n-3

0.91 ± 0.063

0.87 ± 0.040

0.90 ± 0.056

0.88 ± 0.050

0.230

0.097

0.922

C18:3 n-6

0.03 ± 0.011

0.07 ± 0.014

0.06 ± 0.014

0.04 ± 0.018

0.013

0.041

0.958

C20:2

0.04 ± 0.007

0.05 ± 0.006

0.04 ± 0.008

0.04 ± 0.006

0.187

0.941

0.17

C20:3 n-3

0.27 ± 0.022

0.19 ± 0.008

0.25 ± 0.028

0.21 ± 0.013

0.001

0.948

0.083

C20:3 n-6

0.37 ± 0.013

0.37 ± 0.008

0.37 ± 0.013

0.37 ± 0.009

0.867

0.351

0.648

C20:4 n-6

2.46 ± 0.234

1.44 ± 0.052

2.15 ± 0.347

1.75 ± 0.168

< 0.001

0.971

0.007

C20:5 n-3

0.33 ± 0.063

0.22 ± 0.012

0.31 ± 0.061

0.24 ± 0.030

0.044

0.309

0.330

C22:4 n-6

0.50 ± 0.052

0.32 ± 0.012

0.44 ± 0.066

0.37 ± 0.034

< 0.001

0.251

0.032

C22:5 n-3

0.12 ± 0.012

0.10 ± 0.004

0.11 ± 0.009

0.11 ± 0.010

0.065

0.098

0.228

C22:5 n-6

0.14 ± 0.018

0.09 ± 0.006

0.12 ± 0.021

0.10 ± 0.014

0.022

0.028

0.209

C22:6 n-3

0.11 ± 0.027

0.17 ± 0.099

0.13 ± 0.023

0.14 ± 0.026

0.159

0.080

0.889

PUFAc

17.82 ± 0.773

18.49 ± 0.686

19.19 ± 0.411

17.12 ± 0.572

0.450

0.589

0.153

Fatty Acid Indices

16.04 ± 0.701

16.87 ± 0.659

17.42 ± 0.434

15.48 ± 0.527

0.285

0.980

0.258

1.74 ± 0.174

1.55 ± 0.033

1.71 ± 0.151

1.58 ± 0.097

0.076

0.094

0.544

PUFA/SFA

0.54 ± 0.029

0.55 ± 0.023

0.59 ± 0.018

0.50 ± 0.017

0.541

0.589

0.486

n-6/n-3

9.59 ± 0.976

10.86 ± 0.272

10.54 ± 0.931

9.91 ± 0.529

0.161

0.140

0.936

A animal’s with age ranging from 13 – 24 months; B animals with age older than 24 months;

a Saturated fatty acid; b Monounsaturated fatty acids; c Polyunsaturated fatty acids;

e animal age was estimated based on tooth eruption, replacement and wear patter [32];

SEM = standard error of the mean.

Line 99: remove ‘forms’ after concentrate

Removed.

Line 104: animals were shoted not Animals were shot

This was not changed.

Line 109: use italic ‘m. Biceps femoris’ throughout study

Done.

Line 124: remove GLC

Removed.

Line 127: check reference Folch et al. [833

Corrected.

Line 127: repetition ‘fatty acid methyl esters (FAME)’ on line 129

Deleted.

Line 130: remove (FID)

Removed

Results

It's not only results, it contains both results and discussion. Please correct it accordingly

Correction was done.

The English language is poor. Please contact some professionals to improve.

English was enhanced by an English proof-reader.

Discussion should be with proper reasoning and citations. Results are ok but need to present in a better way.

Discussion was improved and new citation were added.

Please correct the whole results and discussion as suggested below

Line 152-154: results are confusing. At first seen, it seems that results are significant. Please correct it and rewrite the sentences ‘Wild boar’s undressed weight was greater at males than females, the average values being 75 vs. 60 kg, with a favourable yield in the case of the males; however, these differences were not statistically significant (P > 0.05)’

Line 152-160: I can't find justification of results. Please justify results with proper reasons and references.

The results were clarified (in table 2 correction were done) and the paragraph was completed and rephrased:

The undressed weight, warm carcass weight, and cold carcass weight of wild boar were higher on males than females, adults vs. sub-adults, with significant differences (P ≤ 0.05). Our findings are in accordance with early preliminary results for the differences found in dressed weight between males (65–108 kg) and females (50–80 kg) of 3–4 years old [27] or other data research summarized by Pedone et al. [26], as cited in Sales and Kotrba's [34] review of the effects of animal age and gender on carcass weights of wild boars hunted in different European countries. These results support the findings of other authors [5, 12, 14, 37, 42], who state that the meat quality of wild boars is determined by the season, available feed, as well as the animals' living conditions and sexual activity.

Table 2. Wild boar’s carcass distribution (mean ±SEM).

Trait

Gender

Age-classe

Effect

Male

(n = 27)

Female

(n = 49)

SubadultsA

(n = 39)

AdultsB

(n = 37)

gender

age-class

gender x

age-class

Undressed weighta (kg)

92.77 ± 9.36

76.76 ± 4.19

71.89 ± 3.77

97.64 ± 7.10

0.03

0.05

0.18

Warm carcass weightb (kg)

74.90 ± 8.29

59.99 ± 3.51

56.49 ± 3.87

78.40 ± 6.35

0.03

0.03

0.13

Cold carcass weightc (kg)

72.01 ± 8.10

57.70 ± 3.52

53.96 ± 3.69

75.75 ± 6.13

0.03

0.04

0.16

Dressing percentaged (%)

77.09 ± 1.33

75.05 ± 0.57

74.79 ± 1.13

77.36 ± 0.75

0.16

0.88

0.73

  • a weight of the freshly shot animal.
  • b without organs but with kidneys, 5 hours after shooting
  • c without organs but with kidneys, 24 hours after slaughtering
  • d cold carcass as a proportion of undressed weight
  • e animal age was estimated based on tooth eruption, replacement and wear patterns [32];
  • A animals with age ranging from 13 – 24 months;
  • B animals with age older than 24 months;
  • SEM = standard error of the mean; ns = P > 0.05; significant = P ≤ 0.05;

Line 172-75: contradictory statements ‘The percentage of protein, ash and water was not influenced by gender and age class (P > 0.05). With the increasing age of the animals, the total lipid content increased significantly (P ≤ 0.001), meat from adults being fatter than meat from subadult wild boars’ please correct it

Correction was done: The percentage of protein, ash and water was not influenced by gender (P > 0.05). Only intramuscular fat content was influenced by age-class, with adults having a significantly (P ≤ 0.001) higher content than subadults, a result that can often be explained by age, feed availability amount, and sexual activity [34].

Line 175-184: very poor justification. Please correct it accordingly

Correction was done: The chemical composition reported by Marsico et al. [20], Strazdina et al. [40] and Lazăr et al. [16] was similar to that found in the present study, in terms of moisture (69–70.5%), protein (20.16–22.92%) and ash content (1.15–1.30%). The influence of gender and age-class on these parameters was not significant (P > 0.05). Our results (over 4.5%) were higher than those reported by the above-mentioned authors, but close to the values obtained by Reka et al. [32] (4.3–6.4% IMF), and in accordance with previous intermediate studies by Postolache et al. [28] (2.65–5.22%) for wild boars hunted in Romania. The results for IMF showed that age-class had a significant impact (P≤0.001), with mean values ranging from 4.52 % for subadults to 7.60 % for adults.

The literature points out that animal diets are reflected in their meat chemical composition, which depends on the season and animal food availability [46]. Moreover, IMF content is influenced by age-class; Neethling et al. (2016) reported that IMF content increases with age, being the last tissue to mature. IMF content differences are mainly a result of animal age and various animal feed intakes: 1.23–4.27% [28], 3.5–5.2% [49], 1.95% [48] or 4.27–14.2% [38].

Conclusion

Although conclusion is drawn from results but poor presentation of results unable me to understand the exact meaning. Rewrite the conclusion again by keeping in view the grammar, sentence structure and technical presentation.

This part was rewritten in order to improve the results presentation by keeping in view the grammar, sentence structure and technical presentation.

According to the above-mentioned findings, the primary unprocessed traits of wild boar carcasses were affected by gender and age-class (P 0.05), with adults, particularly males, weighing more than subadults and females. In terms of chemical composition, total fat content was significantly (P 0.001) influenced by the age-class of the animals, with adults having fatter meat than younger animals. Furthermore, gender and age-class influenced the content of some individual MUFA (C16:1 n-9, C20:1 n-9, C22:1 n-9) and PUFA (C18:3 n-6, C22:5 n-6) fatty acids in the context where total MUFA content was highest in males and adults and PUFA content was highest in females and subadult meat samples.

In summary, these findings suggest that the low intramuscular fat content of wild boar meat is the most advantageous feature that can be used to promote the species. This is due to the fact that sufficient amounts of individual fatty acids in wild boar muscle translate into a health-promoting P/S acid ratio, making it a bio alternative protein resource.

Once again, we appreciate your time on reviewing this manuscript. Hopefully, the revised version of this manuscript is acceptable for this special issue of Animals.

Kind regards,

Dr. Florin LIPSA

Reviewer 3 Report

Remarks:

  1. Figure 1, is illegible, compared to the others, the drawing is not very sharp.
  2. Was the consent of the local ethics committee obtained for the presented research, or was it necessary?

Author Response

Dear Reviewer,

We thank you for reviewing our manuscript and for your suggestions made to improve it.

Please see below the answers (in red) to your comments.

  1. Figure 1, is illegible, compared to the others, the drawing is not very sharp.

Figure 1 was eliminated in order to avoid presenting the same data in both figures and tables. The results from figure 1 are presented in Table 4.

  1. Was the consent of the local ethics committee obtained for the presented research, or was it necessary?

Since the entire study and sampling were performed on dead animals harvested in accordance with the provisions of the hunting law in Romania, such research did not require the consent of the Ethics Committee.

Institutional Review Board Statement was send to the Editor.

Kind regards,

Florin LIPSA

Round 2

Reviewer 2 Report

Authors improved the manuscript according to the suggestions